# Suitable Habitats of *Chrysolophus* spp. Need Urgent Protection from Habitat Fragmentation in China: Especially Suitable Habitats in Non-Nature Reserve Areas

**DOI:** 10.3390/ani12162047

**Published:** 2022-08-11

**Authors:** Peng Wang, Wancai Xia, Enhua Zhou, Yanhong Li, Jie Hu

**Affiliations:** Key Laboratory of Southwest China Wildlife Resources Conservation (Ministry of Education), College of Life Sciences, China West Normal University, Nanchong 637000, China

**Keywords:** *Chrysolophus* spp., potential distribution, habitat fragmentation, conservation

## Abstract

**Simple Summary:**

Wild populations of *C. pictus* and *C. amherstiae* have been decreasing due to habitat fragmentation and long-term uncontrolled poaching. To support the *Chrysolophus* spp.’s conservation, we simulated the potential distribution of the two species in China, calculated the patch fragmentation index of suitable habitats of *Chrysolophus* spp. in nature reserve areas and non-nature reserve areas, and analyzed the habitat status of *C. pictus* and *C. amherstiae* in China. Compared with the previous studies, their habitat areas have been reduced. In addition, most of the suitable habitats were not in nature reserves and were highly fragmented. We offer recommendations for the Chinese government to formulate conservation schemes for the *Chrysolophus* spp. population in the future.

**Abstract:**

Over the past few years, the wild population of *Chrysolophus* spp. has decreased remarkably. Habitat fragmentation is a significant cause for this serious threat to the survival of *Chrysolophus* spp. population. In order to further understand the distribution of potentially suitable habitats of *Chrysolophus* spp., we used the maximum entropy model to predict the potentially suitable habitats of *C. pictus* and *C. amherstiae* in China based on the known distribution. According to the prediction results of the model, we calculated the landscape pattern index to compare the fragmentation of the two species’ potential suitable habitats in nature reserves and non-nature reserves. The results showed that the potentially suitable habitat for *Chrysolophus* spp. only accounted for a small area of China. The suitable habitats for *C. pictus* were mainly in Sichuan, Shaanxi, Hubei, and other provinces, and the model predicts a total area of 359,053.06 km^2^. In addition, the suitable habitats for *C. amherstiae* were mainly distributed in the three-parallel-river area, with a potential total area of 215,569.83 km^2^. The model also showed that there was an overlap of suitable habitats between the two species in the western edge of the Sichuan Basin. Previously, hybrids of the two pheasants have already been found in this same overlapping area predicted by the model. The landscape pattern index showed that in the potentially suitable habitat for *Chrysolophus* spp., the fragmentation of non-nature reserve areas was higher than that of nature reserve areas. The results revealed the distribution of potentially suitable habitats for *Chrysolophus* spp. in China and highlighted that the suitable habitats in non-nature reserve areas were in urgent need of conservation, thereby providing a key reference for the conservation of the *Chrysolophus* spp. population in the future.

## 1. Introduction

A habitat consists of the combined biological and abiotic factors of a landscape that can support animals to enable their survival and reproduction, while a suitable habitat is essential for the survival of entire animal populations [1]. However, with the continuous intensification of climate change and human activities, efforts to ensure the protection and maintenance of global biodiversity are facing great challenges [2]. Although unauthorized hunting has been forbidden by the Chinese government as a wildlife conservation measure [3], the recovery of biodiversity is slow because of economic activities, which lead to habitat fragmentation and degradation [4]. The deterioration and fragmentation of habitats have become the main cause of species and local population extinctions [5].

It has become feasible to obtain point data on species distribution with the rapid development of 3S (Remote sensing, Geography information systems, and Global positioning systems) technology in recent years. Moreover, researchers have used species distribution models (SDMs) in invasive ecology and conservation biology to predict how species and their distributions respond to climate change and other environmental changes [6]. The maximum entropy model (MaxEnt) is a type of species distribution model (SDM), and it was established as a density estimation and species distribution prediction model based on maximum entropy theory [7]. MaxEnt is an algorithm model with a remarkable prediction capability that preserves constraints on environmental data information but has no restrictions on unknown distribution data and missing environmental variables [8]. This model has been identified as useful for determining the distribution changes of wildlife [9,10], and it has been used to demonstrate the dynamic habitat suitability of wild populations under the influence of environmental and human factors [11,12]. It should be noted that this model only requires two sources of data of the target-species occurrences (longitude and latitude) and accommodates a variety of climatic, geographic, interferential, and biological variables [13,14]. Therefore, MaxEnt has been extensively used to assess habitat conservation for threatened species in response to a wide range of variables [15].

Habitat fragmentation refers to the fragmentation of large and once homogeneous habitat patches into isolated and heterogeneous niche patches threatened by natural environment-related or human disturbance [16]. It is a change in landscape spatial pattern and has been considered as one of the main causes of biodiversity loss [17]. Due to the interference of human-derived economic activities, more than 50% of the global original land cover has been altered, resulting in an increasingly serious fragmentation of animal habitats, in turn acting as the main driving force threatening the ecological function of wildlife [18,19]. Of the large ground birds scattered in forests, pheasants rely more extensively on a complete forest ecosystem to obtain the food and habitat needed for survival [20,21]. Habitat fragmentation produces a series of chain reactions for pheasants, increasing the possibility of the local extinction of the population [22,23]. The landscape pattern index can be used to quantitatively describe the characteristics of landscapes’ internal heterogeneity [24]. Due to its good stability for habitat fragmentation analysis, the landscape pattern index has been widely used in the study of regional habitat fragmentation [25,26].

*Chrysolophus* includes two species: the golden pheasant (*Chrysolophus pictus*) and lady Amherst’s pheasant (*Chrysolophus amherstiae*) [27]. *C. pictus* is an endemic mountain forest pheasant species in China that mainly lives in the evergreen broad-leaved forests in southern China [28]. Wild populations of *C. amherstiae* inhabit coniferous and broad-leaved mixed forests and are mainly distributed in China and Myanmar [29,30]. In light of their geographical distribution, feather color, and cytochrome b gene sequence, scholars believe that they are two independent species with a close genetic relationship [31], although there are also hybrids in the wild [32]. *C. pictus* and *C. amherstiae* are the most magnificent species of the pheasant family, and the male bird is world-famous for its showy plumage [33,34]. They are important economic birds, whose feathers are used for decoration, taxidermy, and so on [35,36]. The earliest records of *C. pictus* and *C. amherstiae* can be traced back to the Compendium of Materia Medica during the Ming Dynasty of China (1552 AD to 1578 AD). Uncontrolled poaching of these birds was spurred due to traditional Chinese medicinal theory, which indicates that these two species have great medicinal value [37]. Over recent decades, wild populations of *C. pictus* and *C. amherstiae* have sharply declined in China due to long-term uncontrolled poaching, deforestation, and the loss of suitable habitats [21,38]. Although the two species were listed as a class II-protected species under China’s Wild Animal Protection Law and assessed as Near Threatened (NT) in the Red List of China’s vertebrates [39], their population is decreasing [40,41].

Herein, we used a MaxEnt modeling approach to simulate the spatial distribution of *C. pictus* and *C. amherstiae* in China from 2010 to 2021 and present an assessment of the potential impact with 25 variables on these two species. In addition, we calculated the patch fragmentation index of the suitable habitats of *Chrysolophus* spp. in nature reserve areas and non-nature reserve areas and analyzed the habitat status of *C. pictus* and *C. amherstiae* in China.

## 2. Materials and Methods

### 2.1. Occurrence Records

Occurrence records of *C. pictus* and *C. amherstiae* for this study were compiled from the following four sources during 2010–2021: (I) Infrared camera data. In our previous field survey, we obtained some images of *C. amherstiae* via camera trapping. (II) The vast majority of geographical coordinates of *C. pictus* and *C. amherstiae* were obtained from the global biodiversity information facility (GBIF). (III) Published articles, journals, and some news reports were also resources used as data. Furthermore, (IV) record information was obtained via China bird report. The total occurrence records were 477 for *C. pictus* and 321 for *C. amherstiae*. To minimize errors and avoid having the predicted results far exceed the real distribution due to model overfitting, only one occurrence record was reserved for each grid cell of environmental variables. We excluded those data with unreasonable distributions and repeated records using ENMTools, which can automatically match the grid cell size of environmental variables used for analysis and delete redundant data in the same grid [42]. Ultimately, 182 records of *C. pictus* and 161 records of *C. amherstiae* remained in the final dataset.

### 2.2. Predictor Variable Selection

We selected 25 variables that likely influenced *Chrysolophus* spp.’s habitat selection and movement, including two main categories related to suitable habitat distribution. One category consisted of abiotic factors (i.e., elevation, slope, aspect, land use type, and vegetation types) and the other consisted of biotic factors (i.e., 19 bioclimatic factors and human distribution factors). To avoid the distortion of model estimates due to multicollinearity among the 25 environmental variables, we performed a preliminary screening through MaxEnt and removed variables with a percent contribution of less than 1%. Then, we examined the correlation between 19 bioclimatic factor variables using ENMTools. If |r| > 0.8, this suggested that there was a correlation between variables and the variables with a high percentage of contribution to the MaxEnt Version 3.4.1 (Columbia University, Broadway, New York, USA) model were retained [43]. Ultimately, we chose 4 bioclimatic factors, 3 terrain factors, classification of land use, population distribution, and vegetation type for inclusion into MaxEnt for *C. pictus* (i.e., Bio3; Bio4; Bio17; Ele; Asp; Slo; LUCC; Pd, Vt) and *C. amherstiae* (i.e., Bio4; Bio10; Bio17; Ele; Asp; Slo; LUCC; Pd, Vt) (Table 1).

### 2.3. Suitable Habitat Distribution of Chrysolophus spp. and Niche Differentiation

We modeled the potential distribution of *C. pictus* and *C. amherstiae* in China using MaxEnt Version 3.4.1 (Columbia University, Broadway, New York, USA). Occurrence records of two species and environment variables were fed into this model. We randomly selected 75% occurrence records of *C. pictus* and *C. amherstiae* to train the model prediction and the other 25% of the data were used to test the validity of the model [12]. The model was replicated 30 times to ensure stability. We used the area under the receiver operator curve (AUC), which was threshold-independent, to measure the model’ performance, and the value of AUC ranged from 0 to 1. The distribution probability of sampling sites is higher than that of a random distribution when the value is closer to 1 [44].

We constructed 30 potential distribution models and calculated the average value of different models as the final prediction results. To determine the threshold value, we used maximum training sensitivity plus specificity (MTSS) to conduct binarization processing of the average suitability (occurrence probability) of each grid [45]. Finally, the potential distribution area of *C. pictus* and *C. amherstiae* in China was obtained.

The relative importance of each predictor variable in the model and its impact on the habitat suitability of species was analyzed using the analysis of variable contributions and response curves built into the software. In addition, we compared the similarities and differences in the importance and selection of factors between the two pheasants.

### 2.4. Landscape Analysis

The following 6 indices were used to calculate the potential habitat quality and fragmentation of *Chrysolophus* spp. using FRAGSTATS Version 4.2: Number of patches (NP; NP ≥ 1) reflects the spatial pattern of the landscape and is often used to describe the heterogeneity of landscape, where a high value means high fragmentation. Patch Density (PD; n/100 ha) represents the number of patches per unit area, which reflects the intensity of artificial disturbance to the landscape. With a high PD, landscape fragmentation is more serious. Splitting Index (SPLIT; 0–100) reflects the degree to which the whole landscape is separated and sectioned into several small patches due to interference, with high values indicating that the landscape is more geographically dispersed, the landscape distribution is more complex, and the landscape is more fragmented. The largest patch index (LPI; 0–100) is the proportion of the largest patch area in the whole landscape to the landscape area. The value of LPI determines the abundance of dominant species and internal species in the landscape type. When the maximum patch of the corresponding patch type is very small, the LPI value is close to 0. Patch Cohesion Index (PCI; 0–100) reflects the spatial connection degree of the same type of patches in the landscape. A high value indicates a better spatial connection of the same type of patches, and a poor mosaic. Aggregation index (AI; 0–100) describes the enrichment degree of patch types in the landscape, where low AI value means higher dispersion of patches in the landscape [46].

We utilized the ArcGIS tool Version 10.4 (Environmental Systems Research Institute, Redlands, California, USA) to divide suitable habitats into two parts: nature reserves (including all kinds of nature reserves) and non-nature reserve areas. Six (NP; PD; SPLIT; LDI; PCI; AI) habitat fragmentation metrics were calculated for both nature and non-nature reserve areas in the suitable habitats of *C. pictus* and *C. amherstiae.*

## 3. Results

### 3.1. Model Performance for Chrysolophus spp.

The average value of the model showed an ideal predictive performance after 30 repeated operations, for both *C. pictus* (MTSS = 0.352, AUC = 0.914) and *C. amherstiae* (MTSS = 0.2847, AUC = 0.945).

### 3.2. Potential Suitable Habitat and Niche Differentiation

The prediction results of the MaxEnt model revealed that *Chrysolophus* spp. inhabits a narrow area in China (Figure 1). The potential suitable habitats of *C. pictus* were distributed in 15 provinces of China: Shanxi, Shaanxi, Henan, Hubei, Hunan, Jiangxi, Guangxi, Guangdong, Guizhou, Sichuan, Gansu, Ningxia, Qinghai, Yunnan, and Chongqing (the predicted area of the potential suitable habitat = 359,053.06 km^2^). The maximum patch size of the potential suitable habitat (56,429.9 km^2^, 15.72%) was located in the southeast of Gansu Province, the south of Shaanxi Province, and the west of Henan Province, and the minimum patch size of the potential suitable habitat (0.68 km^2^) was located in the northeast of Qinghai Province, while the potential suitable habitat area of *C. amherstiae* was 215,569.83 km^2^, a smaller potential suitable habitat in China than *C. pictus.* The distribution of the potential suitable habitats for *C. amherstiae* showed a spreading pattern, with the core located in the Hengduan Mountains and the western edge of Sichuan Basin, Yunnan-Kweichow Plateau.

We found that there was an overlap of the potential suitable habitats between *C. pictus* and *C. amherstiae*. The model showed that the overlap area was 17,062.08 km^2^. Moreover, the maximum patch of potential suitable habitat (3923.77 km^2^, 23%) was located on the western edge of the Sichuan Basin.

A variable contribution analysis (Table 2) demonstrated some interspecific differences: altitude (20.1%) was the most important factor for *C. pictus*; the maximum suitability was at an elevation of 1000–2000 m (Figure 2). However, temperature seasonality (41.7%) was the main factor for *C. amherstiae*, and its suitability reached the peak when the temperature seasonality was about 4050 (Figure 3).

### 3.3. Comparison of Potentially Suitable Habitats of Nature Reserves and Non-Nature Reserves

Theoretically, the high number of the NP entail high spatial heterogeneity, which also indicates that the landscape fragmentation is more severe; the number of patches in the non-nature reserves was significantly higher than that in nature reserves for both *C. Pictus* (*Z* = −2.366, *p* = 0.018) (Figure 4) and *C. amherstiae* (*Z* = −2.366, *p* = 0.018) (Figure 5). The patch density (PD) of *C. Pictus* was significantly different between non-nature reserves and nature reserves (*Z* = −2.366, *p* = 0.018), while *C. amherstiae* was not significantly different between non-nature reserves and nature reserves (*Z* = −1.521, *p* = 0.128). The slitting index (SPLIT) of *C. Pictus* (*Z* = −2.366, *p* = 0.018) and *C. amherstiae* (*Z* = −2.197, *p* = 0.028) in non-nature reserve habitats was significantly higher than that of nature reserves, which indicates that the suitable habitats of *Chrysolophus* spp. in non-nature reserves were more seriously fragmented. The largest patch index (LPI) was not significantly different between the non-nature reserves and nature reserves of *C. Pictus* (*Z* = −1.352, *p* = 0.176) and *C. amherstiae* (*Z* = −1.183, *p* = 0.237). The patch cohesion index (PCI) of *C. Pictus* showed that the mosaic of the same type patches in non-nature reserves was poor, and there was a significant difference compared to the nature reserves (*Z* = −2.028, *p* = 0.043). In contrast, *C. amherstiae’s* PCI was not significantly different between the non-nature reserves and nature reserves in the patch cohesion index (PCI) (*Z* = −0.676, *p* = 0.499). The results of the aggregation index (AI) showed that there was no significant difference in the habitat patch dispersion of *C. Pictus* (*Z* = −1.183, *p* = 0.237). However, The concentration of the same type patches in the non-nature reserves of *C. amherstiae* was significantly lower than that in the nature reserves (*Z* = −2.197, *p* = 0.028).

## 4. Discussion

China has abundant and diverse habitat types due to its various climate types and complex topography, so it provides abundant suitable habitats for the survival of different species [12]. However, due to uncontrolled poaching, environmental pollution, climate change, and other human activities, the quality and area of suitable habitats have declined, especially for forests [47,48]. Compared with most other groups of birds, pheasants have a larger body size, a lower reproduction rate, no migration activities, a weaker ability to fly and spread, and a relatively poor ability to escape natural enemies [49]. Therefore, pheasants are more vulnerable to various environmental changes and human activities.

### 4.1. MaxEnt Provided a Well-Predicted Potential Distribution of Chrysolophus spp.

*C. pictus* and *C. amherstiae* are cryptic species that originated in the south of the Qinling Mountains and the Hengduan Mountains, respectively [31]. However, their distributions overlap in some areas, and there are hybrids in the wild [32]. The model also showed us a distribution overlap of *C. pictus* and *C. amherstiae*, and we found that the largest patch in the predicted overlap habitat (30°48′18″ N 103°14′14″ E) was where the presence data derived from the photos taken by Shi [50] in the field were overlayed onto the predicted suitable habitat distribution area (Figure 6). Therefore, our study illustrated that there was a strong correlation between the distribution data of *Chrysolophus* spp. and the predictor variables involved in the construction of the model, and the simulation results of the MaxEnt had high accuracy.

### 4.2. The Existing Suitable Habitat of Chrysolophus spp. Is Shrinking

The results showed that the potential suitable habitat of *Chrysolophus* spp. was much larger than the standard critical cutoff used in species endangerment categories (Species distribution area < 20,000 km^2^) [51]. However, the actual habitat is likely to be smaller because the fundamental niche predicted by the correlative species distribution model is relatively larger than the actual niche [15]. Furthermore, the distribution of species can be constrained by other factors such as inter-species competition and predator and human disturbance, which are not accounted for in the model [8,14].

In our study, we found that most of the suitable distribution areas of *C. pictus* were the same as those in the published literature. In addition, there were some suitable habitats at the junction zone of Hunan, Jiangxi, Guangxi, and Guangdong provinces. According to the literature records [21], *C. pictus* is distributed in a small area in Southeast Tibet, but our model has not found a suitable habitat there for this species. *C. pictus* was monitored by infrared cameras in Guangxi Province and Shanxi Province in recent years, and there were some suitable habitats in these two provinces. However, these patches are too far away from the core distribution areas of the species, and the distance between these patches and the core distribution areas clearly exceeds the dispersal capacity of the species. Therefore, we inferred that there was a geographical isolation among the populations of *C. pictus.* This geographical isolation was due to the gradual decrease in the total area of the optimum habitat, the further reduction of the residual habitat patch area, and the gradual increase of the distance between the patches of each habitat [52]. Furthermore, the loss of habitats increases the chances of inbreeding, which will harm genetic diversity [1].

*C. amherstiae* is a pheasant that originated in the Hengduan Mountains [31], and our model also predicted that the core area of suitable habitat distribution of *C. amherstiae* was in the three parallel rivers of Yunnan’s protected areas (Chin-sha River, Lantsang, and Nujiang) and southwest of the Sichuan Basin. Comparing our predicted distribution of suitable habitats of *C. amherstiae* with known historical habitats [21], we found that the population of *C. amherstiae* in Tibet tended to shrink towards the boundary with Yunnan, and there was no report or article on *C. amherstiae* in Guangxi Province in recent years. Further, temperature seasonality was the most important factor that affected the distribution of *C. amherstiae*. As large terrestrial birds with a poor migration ability, they are weak with respect to coping with habitat changes. Since the seasonal temperature changes affect the carbon assimilation and distribution of plants [53], too small or too large of a temperature seasonality is likely to affect the distribution of *C. amherstiae*. Therefore, it is necessary to reduce the damage of the original plants in *C. amherstiae*’s suitable habitats and avoid changing the vegetation composition in the habitat. Since greenhouse gas emissions continue to cause global warming, the *C. amherstiae* may move to higher latitudes in the future due to its sensitivity to bioclimate. As a major challenge for ecosystems, global warming is predicted to wreak havoc on species with low dispersal capacities [12,54], and some research has predicted that pheasants will migrate to higher altitude or latitude as a response to climate warming [23,55].

### 4.3. It Is Urgent to Protect Suitable Habitats in Non-Nature Reserves

Habitat fragmentation is one of the most important factors contributing to the endangerment of mainland bird species [20]. The main functions of nature reserves are to protect species and habitats as well as to delimit natural resources, wild animals, and plants in areas far away from human interference [56], and they are considered to be among the most effective means of protecting biodiversity and habitats [57,58]. Comparing the habitat fragmentation index of nature reserves and non-nature reserves, the habitat landscape of *Chrysolophus* spp. was extremely fragmented in the non-nature reserve areas, which showed a dense pattern with multiple elements. It is difficult to effectively protect *Chrysolophus* spp. in non-nature reserve areas, where uncontrolled poaching is more serious. In addition, habitats in non-nature reserves are more vulnerable due to human activities such as farmers’ reclamation and land cultivation. Thus, nature reserves play an important role in wildlife protection and management [59]. It is worth mentioning that *C. pictus* was monitored in Jiuwanshan National Nature Reserve in Guangxi Province for the first time. Before that, Ye predicted in their article that there would be almost no suitable habitat for *C. pictus* in Guangxi Province in the future [55]. This further suggests that the establishment of nature reserves and reasonable and effective habitat protection were conducive to restoring the original habitat of the species. By the end of 2019, the nature reserves established in China accounted for 14.8% of the land area [60], and the protection of species and their habitats had achieved remarkable results [61]. However, only some habitats are protected in the nature reserve, and there is still a large area of habitats in non-nature reserve systems, which needs to urgently be protected.

## 5. Conclusions

We estimated the suitable habitat and habitat fragmentation of *Chrysolophus* spp. in China by using the MaXent and Fragstast 4.2. Compared with the previous studies, its habitat area has been reduced. Most of the suitable habitats of *Chrysolophus* spp. were outside of the nature reserves, and this portion of the suitable habitats was seriously fragmented. Although the Chinese government has listed *C. Pictus* and *C. amherstiae* as a class II-protected species under China’s Wild Animal Protection Law and has severely cracked down on the uncontrolled poaching of *Chrysolophus* spp., the species’ protection is still far from sufficient. Accordingly, we propose several implications to help restore the habitat of *Chrysolophus* spp. populations in China: (I) Prioritize the protection of large patches located in the nature reserves. (II) For the suitable habitats of *Chrysolophus* spp. in non-nature reserves areas, it is suggested that the government implement “small protected areas” for protection where conditions permit, and strengthen patrols and monitoring. In addition, poaching should be strictly controlled. (III) Protect the existing landscape and prevent further fragmentation. For those marginal and isolated habitat patches, consider establishing corridors with the surrounding potentially suitable habitats while protecting their integrity. Improve the connectivity of the natural landscape and assess the integrity of habitat patches regularly.

## Figures and Tables

**Figure 1 animals-12-02047-f001:**
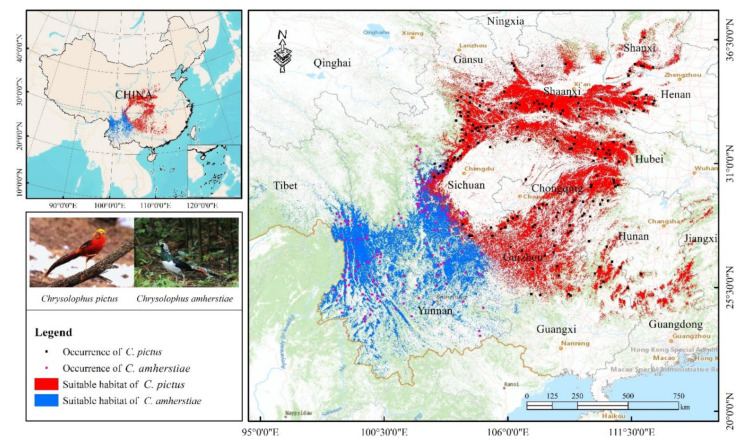
The potential suitable habitats of *Chrysolophus* spp.

**Figure 2 animals-12-02047-f002:**
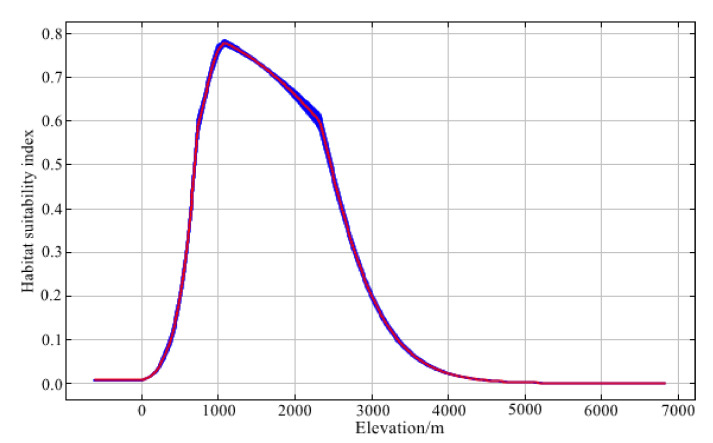
Response curve of *C. pictus* to elevation.

**Figure 3 animals-12-02047-f003:**
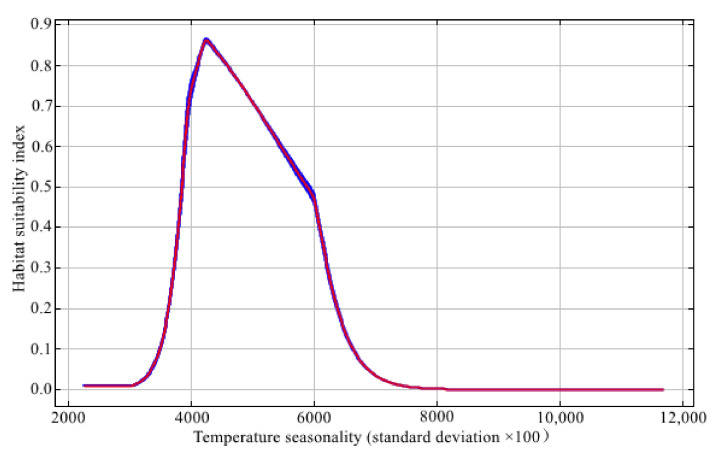
Response curve of *C. amherstiae* to temperature seasonality.

**Figure 4 animals-12-02047-f004:**
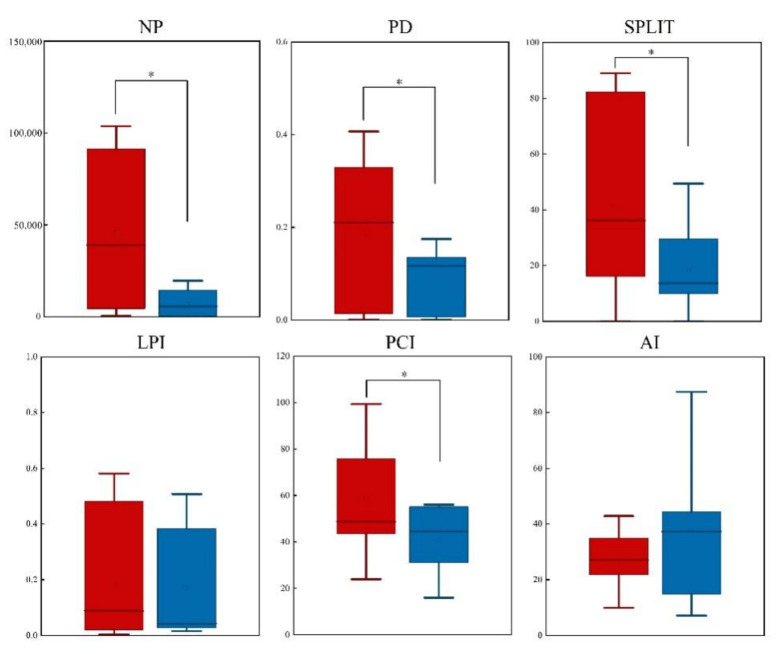
Comparison of potentially suitable habitats of nature reserves (blue) and non-nature (red) reserves of *C. Pictus.* Note: NP: Number of patches; PD: Patch density; SPLIT: Slitting index; LPI: Largest patch index; PCI: Patch cohesion index; AI: Aggregation index. *: *p* < 0.05.

**Figure 5 animals-12-02047-f005:**
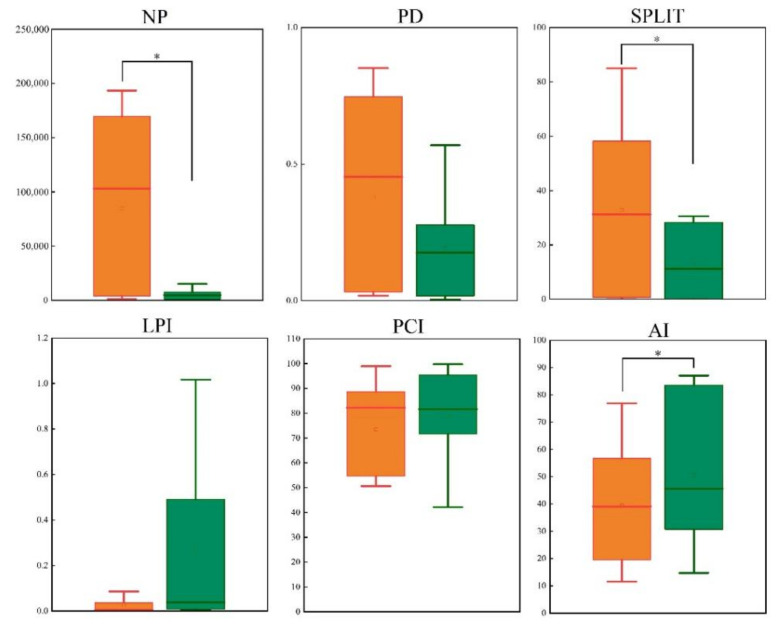
Comparison of potentially suitable habitats of nature reserves (Green) and non-nature (Orange) reserves of *C.*
*amherstiae.* *: *p* < 0.05.

**Figure 6 animals-12-02047-f006:**
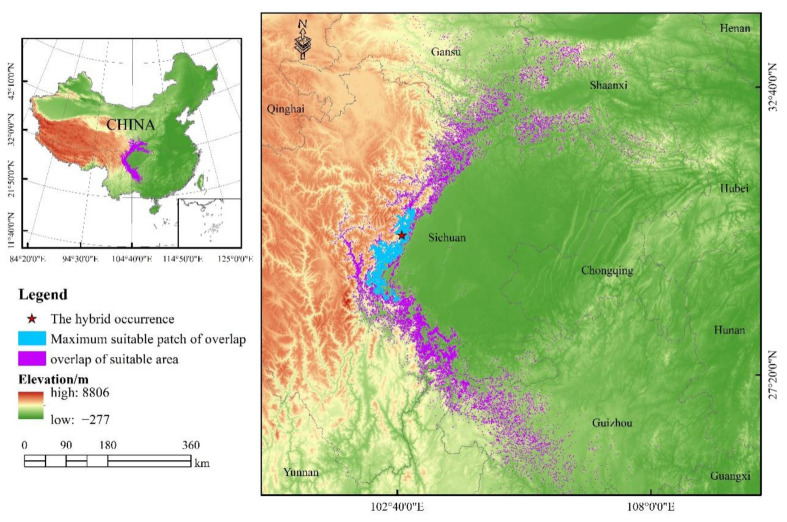
The overlap potential suitable habitat of *Chrysolophus* spp.

**Table 1 animals-12-02047-t001:** Predictor variables for *C. pictus* and *C. amherstiae* and their data sources.

Variables	Description	Source
Bio3	Isothermality	WorldClim database Version 2.0
Bio4	Temperature seasonality	WorldClim database Version 2.0
Bio10	Mean temperature of warmest quarter	WorldClim database Version 2.0
Bio17	Precipitation of driest quarter	WorldClim database Version 2.0
Ele	Elevation	USGS’s Hydro-1K dataset
Asp	Aspect	USGS’s Hydro-1K dataset
Slo	Slope	USGS’s Hydro-1K dataset
LUCC	Classification of land use	National Tibetan Plateau Data Center
Pd	Population distribution	Resource and Environment Science and Data Center
Vt	Vegetation type	Resource and Environment Science and Data Center

**Table 2 animals-12-02047-t002:** Contribution analysis of predictor variables.

Variables	*C. pictus*		*C. amherstiae*	
	Percent Contribution	Permutation Importance	Percent Contribution	Permutation Importance
Bio3	6.5	1.7	-	-
Bio4	14	27.4	41.7	56.7
Bio10	-	-	4.9	17.1
Bio17	8.2	15.1	4.2	4.1
ELe	20.1	34	23.8	10.1
Asp	16.1	11.5	1.3	1.3
Slo	7.2	1.8	3.9	2.8
LUCC	5.2	2.8	3.2	1
Pd	16	4.1	16.1	6.6
Vt	6.7	1.7	0.8	0.4

## Data Availability

The data presented in this study are available from the corresponding author upon request.

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
