# Peer review of "Suitable Habitats of Chrysolophus spp. Need Urgent Protection from Habitat Fragmentation in China: Especially Suitable Habitats in Non-Nature Reserve Areas"

_animals, 2022, doi:10.3390/ani12162047_

Round 1

Reviewer 1 Report

It should be noted that the authors undertook quite an interesting study,
which should have both application and practical importance in the field of
management and protection of endangered species, as a result of various human
activities, and mainly changes in the natural environment.
Taking actions in the field of nature and environmental protection is often
of key importance for the survival of many species, especially in areas
intensively transformed by man. Therefore, the manuscript submitted for
review fits perfectly into the trends of current research in the field
of animal species protection and the search for new solutions in this field.
However, as the authors themselves emphasize in the literature review,
their results coincide with those published by other authors, so in fact
the article does not provide any new, previously unknown information in this
field, and only confirms the existing ones. Also in the literature review,
the authors state that they did not take into account other factors such
as predation pressure or human pressure, and these are, after all, apart
from habitat changes, the key elements affecting the populations of wild
animals. There are many scientific papers backing this up.

After reading the manuscript, I had some remarks: We learn two important things from the introduction. Namely, that hybrids of these species are found in the populations of pheasants studied, so the idea of ​​species protection is not applicable in this case (hybrids do not protect themselves!). The second element is the fact that the cause of the decline in numbers, apart from changes and loss of habitats, is also illegal harvesting, but from the short summary we learn that these are hunting. I am asking the authors not to confuse these two concepts, as they are not the same. Moreover, sustainable hunting harvesting, as a rule, does not contribute to a radical decline in numbers, which is different if it is uncontrolled poaching (!). This must be changed. Line 117-118, what the modification of the data looked like in order to exclude errors, what were the reasons for it, it is necessary to add it, because the text shows that more than half of the collected data was eliminated (why?). It is necessary to add what was the basis of such a decision to eliminate more than half of the data, since they were initially collected and then not used. What was the key to this endeavor and what was it aimed at. Because you get the impression that these data have been manipulated Line 152-153, how the comparison of the similarities and differences in the selection of individual elements by these two species looked like, must be supplemented. The authors in many places, including in the results, state that one of the important factors is the temperature seasonality, which is usually a factor that we do not have much influence on, so what solutions should be proposed, but it is missing in the manuscript. In the conclusions, the authors included three main guidelines, but common sense says that practically all of them in the current era of intensification and even globalization are difficult to implement, especially the establishment of new nature reserves. Although these conclusions are very valuable, I suggest that they are more realistic, perhaps on a smaller scale, but possible to implement.  

I recommend the article for publication after thorough revisions in the places I have described and editorial requirements, as well as more specific and more realistic conclusions in the field of protection

Author Response

Response to Reviewer 1 Comments

Point 1: Namely, that hybrids of these species are found in the populations of pheasants studied, so the idea of species protection is not applicable in this case (hybrids do not protect themselves!).

Response 1: Thank you! This paper describes the habitat protection of the wild pure species of Chrysolophus pictus, and does not involve the protection of their hybrids. There are two descriptions of hybrid species in the article:

(I) Introduction: the purpose of introducing hybrids is to expound the relationship between Chrysolophus pictus and Chrysolophus amherstiae. At present, many scholars think that the two birds are different species whose reproductive isolation has not yet fully formed although there is a natural hybridization phenomenon in the wild.

(II) The occurrence site of field hybrids in published articles are introduced to prove that the distribution predicted by our model is close to the real distribution, which indicates that the model simulation is quite good.

Point 2: The second element is the fact that the cause of the decline in numbers, apart from changes and loss of habitats, is also illegal harvesting, but from the short summary we learn that these are hunting. I am asking the authors not to confuse these two concepts, as they are not the same. Moreover, sustainable hunting harvesting, as a rule, does not contribute to a radical decline in numbers, which is different if it is uncontrolled poaching (!).

Response 2: Thanks to the experts' correction, we have changed the hunting in the article to uncontrolled poaching.

Point 3: This must be changed. Line 117-118, what the modification of the data looked like in order to exclude errors, what were the reasons for it, it is necessary to add it, because the text shows that more than half of the collected data was eliminated (why?). It is necessary to add what was the basis of such a decision to eliminate more than half of the data, since they were initially collected and then not used.

Response 3: Thank you! Almost half of the data were eliminated because the prediction results deviated from the real distribution due to model overfitting. We have supplemented the reasons ( line 119-124) for eliminating some data.

Point 4: What was the key to this endeavor and what was it aimed at. Because you get the impression that these data have been manipulated Line 152-153, how the comparison of the similarities and differences in the selection of individual elements by these two species looked like, must be supplemented. 

Response 4: Thank you! We analyzed the percent contribution of 19 bioclimatic factors once again, and selected those percent contribution of more than 1% to the two species. Then, examine the correlation between 19 bioclimatic factor variables by used ENMTools, and keep the high percent contribution factors. Ultimately, we chose 4 bioclimatic factors, 3 terrain fators, classification of land use, population distribution and vegetation type into MaxEnt for C. pictus (i.e., Bio3; Bio4; Bio17; Ele; Asp; Slo; LUCC; Pd, Vt) and C. amherstiae (i.e., Bio4; Bio10; Bio17; Ele; Asp; Slo; LUCC; Pd, Vt) (supplemented in line 131-140).

Point 5: The authors in many places, including in the results, state that one of the important factors is the temperature seasonality, which is usually a factor that we do not have much influence on, so what solutions should be proposed, but it is missing in the manuscript. 

Response 5: Thank you! The temperature seasonality is obtained by multiplying the standard deviation of temperature seasonality by 100. Some research indicate that temperature seasonality affects the carbon assimilation and distribution of plants. Therefore, the Chrysolophus amherstiae, which is highly dependent on the habitat, will also be affected by the temperature seasonality. We have made supplementary explanations in line 301-307 of the manuscript.

Point 6: In the conclusions, the authors included three main guidelines, but common sense says that practically all of them in the current era of intensification and even globalization are difficult to implement, especially the establishment of new nature reserves. Although these conclusions are very valuable, I suggest that they are more realistic, perhaps on a smaller scale, but possible to implement.  

Response 6: T hanks for the reviewer’s suggestions! Although the establishment of nature reserves is the most effective measure to restore the habitat of species, it is very difficult to implement. We believe that Chrysolophus spp. in non-nature reserve areas are more vulnerable to destructive attacks. Therefore, we’re proposed three suggestions in the manuscript (line 350-358): (I) Give priority to the protection of large patches located in the nature reserves. (II) For the suitable habitats of Chrysolophus spp. in non-nature reserves areas, it is suggested that the government take the form of “small protected area”for protection where conditions permit, and strengthen patrol and monitoring. In addition, poaching should be strictly controlled. (III) Protect the existing landscape and prevent further fragmentation. For those marginal and isolated habitat patches, consider establishing corridors with the surrounding potentially suitable habitats while protecting their integrity. Improve the connectivity of natural landscape and assess the integrity of habitat patches regularly.   

Reviewer 2 Report

This is a generally well conceived study of bird and habitat distribution and valuably assesses the impact of nature reserves on priority species. However, whilst I could understand the majority of the text, the readability of the English could be improved throughout.

There are three areas where I feel improvements could be made to the science. First, variable selection. I did not understand the process by which you reduced from 25 to nine variables and feel this must be more clearly explained. Second I think better justification for your choice of habitat fragmentation metrics is needed - do you need them all or are there two or three which capture key elements of fragmentation? Third, how do you know the magnitude of differences in metrics between nature reserve and non-nature reserve are important? Can you do significance tests or generate confidence limits so you can interpret their degree of overlap?

I think these points are relatively easy to address and think this will be a valuable paper when published.

Author Response

Response to Reviewer 2 Comments

Point 1: First, variable selection. I did not understand the process by which you reduced from 25 to nine variables and feel this must be more clearly explained.

Response 1: Thank you! We analyzed the percent contribution of 19 bioclimatic factors once again, and selected those percent contribution of more than 1% to the two species. Then, examine the correlation between 19 bioclimatic factor variables by used ENMTools, and keep the high percent contribution factors. Ultimately, we chose 4 bioclimatic factors, 3 terrain fators, Classification of land use, population distribution and Vegetation type into MaxEnt for C. pictus (i.e., Bio3; Bio4; Bio17; Ele; Asp; Slo; LUCC; Pd, Vt) and C. amherstiae (i.e., Bio4; Bio10; Bio17; Ele; Asp; Slo; LUCC; Pd, Vt). 

Point 2: Second I think better justification for your choice of habitat fragmentation metrics is needed - do you need them all or are there two or three which capture key elements of fragmentation?

Response 2: Thank you! Landscape pattern index can be used to quantitatively describe the characteristics of landscape heterogeneity. However, it is necessary to select a representative landscape index as the fragmentation analysis in combination with the research purpose. For this reason, we have selected six new indexes for landscape type analysis. The purport of the six index has been added to the corresponding part of the manuscript (line 165-186).

Point 3: Third, how do you know the magnitude of differences in metrics between nature reserve and non-nature reserve are important? Can you do significance tests or generate confidence limits so you can interpret their degree of overlap?

Response 3: Thanks to the reviewer' correction! We performed Wilcoxon signed-rank test on the landscape index values calculated by FRAGSTATS V.4.2, and compared the fragmentation of non-nature reserve areas and nature reserve areas (line 227-247).
